# One-day repeat pass interferometry highlights the role of temporal baseline on digital elevation models retrieved from Sentinel-1

Andreas Braun[1]

[1] Institute of Geography, Department of Geosciences, University of Tübingen, Tübingen, 72070 Germany

*Correspondence to*: Andreas Braun (an.braun@uni-tuebingen.de)

**Copyright statement**

*To be included by Copernicus.*

**Abstract.** Digital Elevation Models (DEMs) derived from Synthetic Aperture Radar (SAR) interferometry are a key data source for numerous geospatial applications, from hydrological modelling to environmental monitoring. The launch of Sentinel-1C in late 2025 introduces a new sensor into the Sentinel-1 constellation. This study evaluates the vertical accuracy of DEMs generated from interferometric image pairs acquired during the satellite's calibration phase. The analysis uses a set of image pairs with temporal baselines of 1, 6, and 12 days, over a test site in Angola, validated against ICESat-2 elevation measurements. The workflow includes interferometric processing, coherence assessment, and statistical error evaluation. Results indicate high accuracy for the 1-day pair (RMSE ≈ 14.7 m) and moderate degradation for the 6-day pair (RMSE ≈ 16.4 m), but a pronounced loss of accuracy for the 12-day pair (RMSE ≈ 49.4 m), primarily linked to coherence loss in vegetated areas. Coherence and elevation error distributions reveal clear land cover and slope dependencies, with lower performance in forested and steep terrain. These findings should be regarded as indicative due to the limited number of suitable image pairs for the calibration phase. However, this early assessment provides an important reference point for future Sentinel-1A/C DEM generation studies, informing both methodological refinement and application planning in SAR-based topographic mapping.

## 1. Introduction

Digital elevation models (DEMs) are an essential data source for the analysis of terrain, geomorphologic and hydrological processes and risks and climate-induced changes of terrestrial ecosystems (Moore et al., 1991; Schillaci et al., 2015; Guth et al., 2021). Besides photogrammetric approaches and aerial campaigns, synthetic aperture radar (SAR) missions, such as the Shuttle Radar Topography Mission (SRTM) or TanDEM-X are have set new standards to retrieve consistent and high-resolution elevation data over land, especially at global scale (Farr et al., 2007; Rizzoli et al., 2017). They are based on the interferometric principle which uses the phase difference between two spatially or temporally distinct acquisitions to measure surface heights relative to the sensor (Zebker and Goldstein, 1986; Madsen et al., 1993; Bamler and Hartl, 1998).

The launch of the Sentinel-1 mission within the Copernicus Programme by the European Space Agency (ESA) marked the beginning of a new era of radar observations, as it delivered radar imagery for the first time that was openly available for research, public and commercial purposes, at regular intervals and with high spatial resolution and global coverage (Torres et al., 2012). The continuity of the mission was ensured by a series of three nearly identical Sentinel-1 satellites (S-1A, S-1B and S-1C), which were launched in 2014, 2016 and 2024 respectively (Torres et al., 2021). This has provided consistent and seamless coverage for over a decade, enabling the development of both dense and long-term environmental monitoring and change detection applications (Confuorto et al., 2021; Wagner et al., 2021; Monti-Guarnieri et al., 2022).

However, Sentinel-1's mission and sensor design mainly favor differential interferometry (DInSAR) targeting the precise measurement of surface deformation due to earthquakes or mass movements (Funning and Garcia, 2018; Mantovani et al., 2019; Crosetto et al., 2020), rather than the derivation of digital elevation models. The latter is mainly prevented by the combination of the wavelength of the C-band sensor and the 12-day repeat cycle (or 6 days during phases of parallel operation of S1-A and S-1B), which causes the temporal decorrelation of most natural surfaces over short periods (Yagüe-Martínez et al., 2017; Kellndorfer et al., 2022), as well as the predominantly small orbital tube of the mission primarily designed to detect surface displacements by differential interferometry (Barat et al., 2015). While this prevents the exploitation of high-quality phase information for the derivation of DEMs, various studies have indicated the potential of topographic mapping when the image pairs and the study area meet the necessary preconditions, which mainly include short temporal baselines (the time between the acquisition of the first and second image of the pair), large perpendicular baselines (the distance between the orbit positions of both satellites at the time of their acquisition), and little vegetation cover (Braun, 2021).

As a consequence of the failure of S-1B in late 2021, the launch of its successor S-1C was highly anticipated and realized in 2024, allowing for a return to the 6-day repeat cycle between both operating satellites. Between January and March 2025, an initial calibration and validation phase of Sentinel-1C featured acquisitions with exceptional short temporal baselines of one day to the existing S-1A acquisitions for selective imagery over Europe, Greenland, and Africa. This unique constellation allowed to systematically investigate the impact of the temporal baseline on DEM quality and thus quantify key limitations of the C-band.

In this work, digital elevation models from image pairs taken 1, 6, and 12 days apart are analyzed comparatively and evaluated with respect to different land cover and topographical conditions. The aim is to systematically determine the impact of the temporal baseline on the quality of Sentinel-1 DEMs in order to better understand the sensitivity of the C-band and derive reliable information for the design of future radar missions.

## 2. Data and Methods

The study area was selected based on a list of criteria in order to isolate the influence of the temporal baseline. These were:

a)  Sentinel-1C Single-Look Complex (SLC) products in Interferometric Wide Swath (IW) mode acquired between 07 January 2025 and 10 March2025

b)  Availability of a complementary Sentinel-1A image from the same relative orbit taken 1 day apart

c)  Availability of image pairs at baselines of 6 and 12 days for reasons of comparison from the same relative orbit

d)  Perpendicular baselines of comparable length for all selected image pairs, ideally larger than 150 m to allow a proper description of the topographic fringes (Ferretti et al., 2007).

e)  Area contains landscape with pronounced topographic variation and ideally different types of landcover

Systematic queries were performed in the Copernicus Dataspace Ecosystem (CSDE) to ensure criteria a) to c). As it turned out, the acquisitions over Europe were limited to Sicily and those over Africa originated from the same relative orbit (58), as shown in Figure 1A. Only one frame along this orbit fulfilled the criteria d) and e), mainly because of strong variations in the perpendicular baseline which would bias the actual effects of temporal decorrelation which are of interest in this study. As shown in Figure 1B, this frame lies in the south of Angola and features a heterogeneous land-use mosaic with cropland (dryland and irrigated fields), patches of shrub/grassland, and compact settlement zones along major transport corridors, while more natural vegetation persists on steeper slopes and ridgelines. The topography is moderate with an average altitude of 1330 m above sea level, ranging between 1250 m and 1450 inside the analysed area. 90% of all slopes are below 5° predominantly ranging from Northwest to Southeast, especially in the eastern part of the study area which is covered by trees (Figure 1C). Geologically, the landscape consists of gently to moderately dissected hills with bedrock exposures on upper slopes and colluvial–alluvial deposits in valley floors, yielding thin soils on crests and deeper profiles on footslopes and floodplains. The hydrosphere is characterized by intermittent streams and small impoundments, with groundwater primarily hosted in alluvial fills and weathered horizons; during the winter–spring acquisition window, soil-moisture levels are seasonally elevated (Huntley, 2019).

Table lists the image pairs which were identified as suitable for this study, as well as their temporal ($B_{temp}$) perpendicular baselines ($B_{perp}$) and the resulting height of ambiguity (HoA) which defines the elevation distance which is covered by one phase cycle in the interferogram. It shows that the first three image pairs are comparable with respect to the acquisition geometry. These three pairs (1-3) are the ones that the subsequent analyses are based upon. A fourth pair (pair 4) was additionally analyzed to double-check the typical behavior of a standard 12-day Sentinel-1 repeat cycle under comparable environmental conditions (a maximum perpendicular baseline of only 148 m was available), while acknowledging differences in acquisition geometry that preclude a direct quantitative comparison. Additionally, an analysis of rainfall data of the ERA5 dataset (C3S, 2018) dataset showed that there were no significant rainfall events between all pairs so that quality differences in the derived DEMs can mainly be assigned to the temporal baseline. Full scene identifiers are provided in the appendix to foster reproducibility and transparency.

All input data were processed in the ESA Science Toolbox Exploitation Platform (SNAP) as described in (Braun, 2021) which included the steps summarized in Table 2.

To assess the quality of the generated DEMs of each image pair, the Copernicus Global Digital Elevation Model GLO-30 (ESA, 2022) was used as it provides high accuracy elevations at global coverage with an absolute vertical accuracy of <4m and a relative vertical accuracy of <2m for slopes <20% (Airbus, 2022). As it originates from data of the bistatic TanDEM-X mission, it can be considered fully independent from the DEMs produced in this study (Marešová et al., 2021). In this study, it was used for visual comparison of the generated DEMs (Sect. 3.2) and for calculation of terrain slope as a potential influencing factor on the InSAR DEM quality. However, to also employ a non-interferometric reference, measurements of the altimetric ICESat-2 mission (Neuenschwander et al., 2023) were used as a second quality indicator. The mission produces discrete laser footprints on the ground with a nominal diameter of around 14 m at intervals of around 90 m along the flight path (Magruder et al., 2021) of which 12.727 fall within the study area for the period between January 2024 and March 2025. At these locations, surface elevation measurements ("terrain best fit") at sub-meter accuracy (Zhu et al., 2022) were sampled as the absolute height reference to be used for the computation of accuracy metrics in the following.

## 3. Results

### 3.1. Coherence

In a first step, interferometric coherence is computed as the magnitude of the normalized complex cross-correlation between two co-registered SAR SLC images over a local window. It quantifies the stability of the scattering phase between the acquisitions and ranges from 0 (no correlation) to 1 (perfect stability), and therefore serves as an early indicator for the final DEM quality of each pair (Martone et al., 2012). Figure 2 shows the coherence maps retrieved from the three image pairs as well as their histograms and raster statistics. The maps show bright areas with high coherence especially for pair 1 ($B_{temp}$ 1 day) in areas with less vegetation cover, mainly along the river stream and an average of 0.468 over the entire image. In comparison, strong coherence is less frequent and less spatially connected in pair 2 ($B_{temp}$ 6 days) and also slightly lower at average (mean 0.466), although their histograms are widely identical. A strong decrease can be observed between pair 2 and pair 3 ($B_{temp}$ 12 days) which is largely decorrelated except for areas in the center (mean 0.346). This shows the impact of temporal decorrelation over vegetation which is a common problem in radar interferometry (Zebker and Villasenor, 1992). As a supplementary robustness indicator, the equivalent number of looks (ENL) is calculated and considered, which describes the effective number of independent looks and thus the variance reduction through multi-looking (Jong-Sen Lee et al., 1994). Accordingly, a higher ENL represents lower estimation variance (Gierull and Sikaneta, 2002). The ENL is almost identical for pair 1 (2.246) and pair 2 (2.254), but drops to 1.452 for pair 3, indicating significantly poorer phase estimation precision at 12 days. Overall, the coherence analysis supports the expectation that a 1-day repeat provides noticeably more favorable conditions for height derivation while temporal decorrelation predominates with the pair of 12 days. Coherence is analyzed at more detail in Sect. 3.5 and 3.6.

## 3.2. Interferograms and Digital Elevation Models

Interferograms of all pairs are presented in Figure 3 together with the DEMs resulting from the processing outlined in Table 2, as suggested by Braun (2021) to identify potential sources of error at an early stage. In contrast to coherence, interferograms provide direct information about phase quality and the achievable level of detail. Additionally, all DEMs were overlaid with hill shading to better highlight subtle differences. For reasons of comparison, the Copernicus Global Digital Elevation Model (GLO-30) is additionally displayed at the bottom. Pair 1 ($B_{temp}$ 1 day) shows high phase quality with clearly pronounced fringes. As indicated in Figure 2, phase noise is limited to areas of low coherence. However, a seamline is clearly visible along the border between bursts 2 and 3 in the lower part of the area as a processing artefact after the debursting process (highlighted by a dashed black line). This seamline comes with strong phase jumps and is not present in any of the other pairs and is most likely a consequence of the experimental nature of the Sentinel-1C acquisitions, for which the calibration quality was explicitly stated to be degraded (Hajduch, 2025). This problem could not be solved using adjusted processing parameters, and it represents an intrinsic bias that unfortunately affects the final results, primarily by overestimated heights in the lower central area of the data. Yet, the produced DEM well aligns with the reference data of GLO-30 with only smaller height deviations and the bias caused by the aforementioned phase jumps. The interferogram of pair 2 ($B_{temp}$ 6 days) is nearly identical and has slightly larger phase noise, but with less systematic height errors because it is not affected by the phase jump (despite the involvement of Sentinel-1 data from 20 April 2025). In comparison to pair 1, it shows a more consistent terrain surface. Pair 3 ($B_{temp}$ 12 days) shows clearly higher amounts of phase noise as a consequence of temporal decorrelation which lead to lower DEM quality because of subsequent unwrapping errors in areas of non-resolvable phase information (Yu et al., 2019b). In the resulting elevation model, this manifests itself in local artifacts and a loss of fine-scale relief detail, also strongly overestimated elevations in the southern part of the area.

Looking at all interferograms, it can be stated that the similar perpendicular baseline leads to a comparable height of ambiguity and thus similarly dense fringe patterns, which are necessary for a precise description of the relief. The differences in quality can therefore be attributed to the temporal baseline and the systematic error, not the acquisition geometry.

## 3.3. Error metrics

The following error metrics were computed based on the reference surface heights retrieved from the ICESat-2 mission (Sect. 2), the elevations of the three analyzed image pairs ($z_i$), and their difference ($\Delta_i = z_i^{ref} - z_i$), with $n$ as the number of observations:

Root Mean Square Error (RMSE, Eq. 1): The square root of the mean of the squared differences between estimated and reference elevations. It quantifies the overall magnitude of elevation errors, giving more weight to larger deviations, and is useful for assessing the general accuracy of DEM products.

$$RMSE = \sqrt{\frac{1}{n}\sum_{i=1}^{n}(\Delta_i)^2} \tag{1}$$

Normalized Median Absolute Deviation (NMAD, Eq. 2): Computed as 1.4826 times the median absolute deviation from the median of elevation differences. It is robust against outliers and is particularly suitable for characterizing the typical vertical error in DEMs when the error distribution is non-normal.

$$NMAD = 1.4826 \cdot median(|\Delta_i - median(\Delta)|) \tag{2}$$

Linear error with 90% confidence (LE90, Eq. 3): Calculated as 1.6449 times the RMSE, represents the error level below which 90% of elevation differences are expected to fall, assuming a normal distribution, and is a common metric in geospatial accuracy standards.

$$LE90 = 1.6449 \cdot RMSE \tag{3}$$

Mean Bias (Eq. 4): The arithmetic mean of the elevation differences. Indicates whether the DEM has a systematic tendency to overestimate or underestimate elevations relative to the reference. Its range is indicated by red dashed lines in Figure 4 which displays histograms of the error ($\Delta_i$) of the DEMs from the three image pairs.

$$Mean\ Bias = \frac{1}{n}\sum_{i=1}^{n}\Delta_i \tag{4}$$

Mean Absolute Percentage Error / Relative Height Residual (MAPE, Eq. 5): The mean of the absolute elevation differences divided by the absolute reference elevations. It is expressed as a percentage to allow for comparison between areas of different terrain elevations (Willmott and Matsuura, 2005).

$$MAPE = \frac{100}{n}\sum_{i=1}^{n}\frac{|\Delta_i|}{\left|z_i^{ref}\right|} \tag{5}$$

The results of the accuracy assessment are shown in Table 3 and show that both RMSE and NMAD noticeably increase with longer temporal baseline, especially between pair 2 ($B_{temp}$ 6 days) and pair 3 ($B_{temp}$ 12 days), confirming that loss of coherence and associated phase noise are non-linear with respect to the temporal baseline. This is also confirmed by several studies on DEM generation with InSAR which report that after a certain coherence threshold is crossed, unwrapping errors and phase decorrelation produce disproportionately large height errors (Braun, 2021). Comparing robust and non-robust metrics, the table shows that NMAD and RMSE are similarly low for pair 1 and pair 2, suggesting that the error distribution is relatively symmetric and not strongly affected by outliers. These measures are also similar for pair 3, but three times larger in general, which indicates that the entire error distribution has shifted to higher variability rather than being dominated by a few extreme outliers. Since LE90 is just 1.6449 × RMSE here, its behavior mirrors RMSE exactly. For comparison, the

GLO-30 has an RMSE of 3.496 m. The mean bias increases from 1.48 m (1 day) to 3.61 m (6 days) and 20.40 m (12 days). This is also visible by the error histograms in Figure 4 which show that errors are largely symmetric for pair 1 and skewed to the right in pair 2 and 3. MAPE values are small for pairs 1 and 2 (~0.87-0.89%) but triple for pair 3 (~2.98%), which is in turn consistent with a proportional error growth. Because MAPE is scale-free, this suggests that the quality degradation is relative to terrain magnitude, not only in absolute terms. Low MAPE values in combination with high NMAD values in pair 3 may indicate that large deviations are concentrated in steep or high terrain while high MAPE values with high NMAD values point to more widespread degradation. This is further analyzed in Sect. 3.5.

## 3.4. Bias analysis

To provide a broader context for the significant differences between pairs 2 and 3, and to analyze whether the deterioration in DEM quality is solely due to a higher temporal baseline or if other factors are also contributing, a planar trend analysis was performed on the elevation residuals, which were calculated by subtracting each pair's elevation values from those of the ICESat reference heights. This was done by fitting a first-order polynomial surface of the form $z = a + bx + cy$ to these residuals using least-squares regression , where $x$ and $y$ denote UTM Easting and Northing coordinates (Brovelli et al., 1999). The fitted plane was subsequently removed from the residuals, and selected error metrics were recomputed on the detrended data (Table 4). This approach isolates long-wavelength artefacts such as residual orbital errors, large-scale atmospheric phase contributions, or unwrapping reference effects (Hanssen, 2001; Wu and Madson, 2024). The analysis reveals that pair 3 exhibits a pronounced large-scale ramp, with gradient magnitudes of approximately 0.90 m/km in East–West direction and 0.98 m/km in North–South direction, exceeding those of the 1- and 6-day pairs by more than an order of magnitude. Over the spatial extent of the study area, this corresponds to systematic elevation offsets on the order of several tens of metres, consistent with the observed bias and error dispersion. As shown in Table 4, removing this ramp from pair 3 reduces the NMAD from 49.3 m to 22.4 m and the LE90 from 81.3 m to 51.2 m. This demonstrates that the strong degradation in DEM quality is dominated by systematic long-wavelength errors rather than by random noise or temporal decorrelation alone. In contrast, ramp removal had only a marginal effect on the error metrics of pairs 1 and 2 supporting the absence of pronounced large-scale systematic error components, except for the phase jump demonstrate in pair 1.

To add more evidence to these numbers, error metrics of pair 4 (Table 1) were analyzed to provide qualitative context from comparable temporal baselines. Although this pair exhibits a substantially smaller perpendicular baseline (~150 m) than pair 3 (~300 m), the comparison reveals markedly different error characteristics: While pair 3 shows a pronounced systematic vertical offset, pair 4 exhibits a different bias magnitude and dispersion pattern (RMSE: 43.628 m, NMAD: 57.673 m, Mean Bias: 15.683 m). This divergence indicates that the strong degradation observed in pair 3 is not a consistent feature of 12-day temporal baselines but rather reflects pair-specific error behavior influenced by acquisition geometry and phase referencing.

### 3.5. Impact of Terrain

Terrain slope was computed based on the GLO-30 DEM and added to all sample points used in the previous sections to analyze if topography has an impact on coherence and height errors. As large proportions of the study area are predominantly flat (see Figure 3) and only small factions show high slope angles, four classes (0 to 2.5°, 2.5 to 5°, 5 to 7.5°, and >7.5°) were defined for this analysis. Figure 5Figure 5 shows box plots of coherence values of all three analyzed image pairs grouped by the defined slope classes.

Across all three image pairs, the median coherence decreases with increasing slope with the strongest decline appearing in the >7.5° class. Interestingly, the strongest decrease in coherence is observed in pair 1 ($B_{temp}$ 1 day), with a median decrease from 0.48 in flat terrain to 0.40 in the steepest class. For pair 2 ($B_{temp}$ 6 days), absolute coherence values are slightly lower throughout all classes and the decline with slope persists, but less steep. Median coherence is already much lower (0.34) in flat terrain for pair 3 ($B_{temp}$ 12 days) and decreases to 0.30 while distributions broaden. However, these comparisons have to be interpreted with care because statistics of the slope classes are based on very different sample sizes (n=8338, n=3871, n=471, and n=46) as a consequence of the equal interval classification. Yet, trends are consistent throughout all three pairs, and it can be stated that steeper terrain leads to lower coherence in general and thus to a poorer data quality for the subsequent interferometric processing.

In a next step, the elevation differences (Sect. 3.3) were disaggregated by the defined slope classes and plotted as shown in Figure 6. Similar to the coherence statistics, the lowest slope class appears to contain the largest variance at first glance, but this can again be attributed to the larger sample size in flat terrain (n=8338). Median elevation differences lie around 0 m through all classes, and interquartile ranges (IQR; representing the center 50% of all sampled elevation differences) are nearly identical across the first three classes, ranging from around -5 to around +10 m. Also, the whiskers, representing the 5% and 95% percentiles, have largely similar ranges from around -22 to +25 m. The class with the highest slopes (>7.5°) seems aligned with these numbers but should be interpreted with care due to the small number of samples (n=46). For pair 2 ($B_{temp}$ 6 days), positive deviations occur more frequently as compared to pair 1 ($B_{temp}$ 1 day), yet the median height error remains within -1 and +1 m in all classes. Whisker lengths are comparable to pair 1, indicating robust, largely relief-independent accuracy. For pair 3 ($B_{temp}$ 12 days), the distributions broaden markedly, with IQR roughly from -10 to +50 m and clearly longer whiskers. Occasional outliers appear, particularly at steeper slopes, pointing to a notable loss of elevation quality as consequences of local layover and shadow effects. Overall, no systematic median bias across slope classes for pair 2 can be identified, indicating comparable quality for pairs 1 and 2 ($B_{temp}$ 1 and 6 days), while accuracy primarily degrades between 6 and 12 days. The predominance of gentle slopes strengthens the statistical reliability of the first two classes, whereas conclusions for >7.5° remain tentative due to small sample sizes. Accordingly, the deterioration at pair 3 ($B_{temp}$ 12 days) could be interpreted as the combined effect of increased phase noise and higher unwrapping susceptibility in complex

terrain, which broadens the error distributions. This pattern is consistent with coherence analysis and underscores the value of short repeat intervals for high-quality DEM generation.

## 3.6. Impact of Land Cover

To assess if land cover, which is strongly linked to different backscatter mechanisms of surfaces, affects the quality of interferometric radar products, both coherence height errors were overlaid with land cover classes at the sample points. These were retrieved from the ESA WorldCover dataset (Zanaga et al., 2022). Figure 7 shows boxplots of coherence for all three image pairs grouped by the main classes (Tree Cover, Shrubland, Grassland, Cropland, Herbaceous Wetland). The overall trend from Sect. 3.1 is confirmed: coherence decreases with increasing temporal baseline across nearly all classes. Throughout all classes, Tree Cover exhibits the lowest coherence (0.45, 0.44, 0.36) because of the large proportions of volume decorrelation (Kellndorfer et al., 2022) while Grassland contains the highest medians (0.64, 0.54, 0.45) as a result of surface scattering dominance (Stiles et al., 2000). All other classes show indifferent statistics over the three analyzed pairs. All classes have the highest coherence in pair 1 ($B_{temp}$ 1 day), with median values above 0.5 except for Tree Cover. In contrast, coherence in pair 2 ($B_{temp}$ 6 days) drops markedly for the classes Cropland and Herbaceous Wetland because temporal decorrelation occurs already within a few days (Mestre-Quereda et al., 2020). Grassland declines moderately and Tree Cover remains low and largely unchanged, consistent with pre-existing volume decorrelation. For pair 3 ($B_{temp}$ 12 days) median coherence falls below 0.4 in all classes with Tree Cover decreasing further and Cropland becoming the lowest coherence class. Grassland retains the highest coherence in comparison but remains well below its pair 1-level. These observations align well with expectations from volumetric and temporal decorrelation: forested and agricultural surfaces decorrelate more strongly than grasslands (Kellndorfer et al., 2022). The transition from 6 to 12 days produces a cross-class drop in coherence that is evident even in structurally simpler surfaces such as Grassland.

Figure 8 presents boxplots of the elevation deviation by ESA WorldCover class. At first glance, the differences between pair 1 ($B_{temp}$ 1 day) and pair 2 ($B_{temp}$ 6 days) are generally small and only Cropland shows a tendency toward positive deviations (95% percentile increases from 21.1 to 38.6 m), confirming the quick decorrelation of the signal as explored above. For pair 3 ($B_{temp}$ 12 days), height uncertainty increases markedly across all classes: interquartile ranges widen throughout all land cover classes, and all medians shift to positive values, indicating systematic overestimation. The effect is strongest for Tree Cover (IQR between -10.9 to +57.5 m; median +25.4 m), followed by Shrubland. Herbaceous Wetland exhibits the strongest overall positive shift. This increase in elevation errors can be attributed to combination of temporal decorrelation and greater unwrapping susceptibility at 12 days which introduces positive biases, particularly in volume-scattering or dynamic classes (Forests, Shrublands, Wetlands). The relative stability up to 6 days and the pronounced degradation by 12 days is consistent with the coherence analysis.

## 4. Discussion

The systematic comparison of interferometric pairs of similar perpendicular baseline showed the role of the temporal baseline as a critical factor controlling DEM accuracy. The presented results show a highly non-linear degradation of coherence and elevation precision with increasing time separation between acquisitions. The decrease in DEM quality from 6 days to 12 days baseline was far more pronounced than from 1 day to 6 days, indicating a threshold beyond which C-band temporal decorrelation dominates the error budget. This suggests that once the temporal baseline extends beyond about a week, phase coherence over vegetated terrain decreases and unwrapping errors emerge, leading to disproportionately large height errors. This observation is consistent with previous studies of InSAR DEM generation which report that after a certain coherence loss, the phase information becomes too noisy to recover reliable heights (Wu and Madson, 2024). Only few existing studies quantified this effect: Braun (2021) reported an average decrease of coherence of -19.2% between Sentinel-1 image pairs with temporal baselines of 6 and 18 days. Yan et al. (2025) compared multiple image pairs separated by 6 and 12 days and found a decrease in standard deviations from around 8.7 m to 21.3 m (6 day pairs) to 36.2 m to 67.9 m (12 day pairs). Zyshal et al. (2021) also compared error metrics of pairs of different temporal baselines, underlining the strong drop in DEM quality between 6 days (RMSE 0f 32.9 m) and 12 days (RMSE of 158.9 m). This study supplements these presented figures with additional error metrics for Sentinel-1, even if limited to a very specific study region and observation phase: RMSE, NMAD remained relatively low and comparable for 1-day and 6-day pairs but then tripled when the baseline extended to 12 days. Correspondingly, coherence values dropped dramatically for the pair of 12 days, but a more detailed decomposition of elevation errors was required to distinguish systematic effects from random elevation noise and systematic bias. Such a bias could potentially stem from unmodeled atmospheric phase delay gradients or residual orbital errors that were not canceled out, as well as the cumulative effect of unwrapping ambiguities (Devaraj and Yarrakula, 2020; Hanssen, 2001). After removing this ramp from the elevation differences for analytical purposes, NMAD was reduced by more than 50 %, and LE90 by nearly 40 %, underlining additional effects on DEM quality deterioration besides larger temporal baselines. The important implication is that, unlike random noise, a systematic bias can be identified and potentially corrected if its source is understood (Fattahi and Amelung, 2013; Danudirdjo and Hirose, 2015; Liu et al., 2020). In this case, correcting the ~20 m bias in pair 3 (for example, by using reference elevation data or atmospheric correction models) would bring its accuracy considerably closer to the shorter-baseline results. This underlines the value of characterizing and mitigating biases in interferometric DEMs an aspect that becomes increasingly important for longer temporal baselines. Results on coherence show that temporal decorrelation (especially over vegetated areas) is an important driver of accuracy loss in C-band DEMs (Kolecka and Kozak, 2014; Morishita and Hanssen, 2014).

Unfortunately, the utility of the results are limited by the fact that Sentinel-1C's experimental status introduced notable data quality issues (Hajduch, 2025). Sentinel-1C imagery used in this study was acquired during its calibration/validation phase and had explicitly degraded calibration quality. In practice, this meant that precise orbital information was unavailable and burst synchronization with Sentinel-1A could not be guaranteed. These factors likely contributed to the seamline artifact

observed in pair 1 (1-day baseline), where a discontinuity with abrupt phase jumps led to locally inflated elevation values. This issue could not be eliminated through processing tweaks, indicating an intrinsic bias in the Sentinel-1C data that propagates into the DEM as systematic height errors. It cannot be quantified to what extent the DEM quality of pair 1 would have exceeded that of pair 2 if this systematic error had not occurred due to back geocoding, but the errors would have been smaller in any case. Thus, both show similarly high accuracies and the final outperformance of 1-day baselines remains partly undetermined.

There is an inherent trade-off between temporal and geometric baselines in InSAR DEM generation. Short revisit intervals minimize temporal decorrelation, preserving coherence, but they often coincide with smaller perpendicular baselines, which degrade the vertical resolution of the DEM (a small baseline yields a large height-of-ambiguity). Conversely, a large perpendicular baseline improves the sensitivity to topography (lowering the height-of-ambiguity) but can come at the cost of reduced coherence if the acquisition times are farther apart or the imaging geometry changes significantly (Yu et al., 2021). In this study, the three image pairs had similar perpendicular baselines (~307-386 m) by experimental design, so height sensitivity was comparable. This ensured that differences in DEM quality are attributable mainly to temporal decorrelation. Generally, though, mission planners must balance these factors: an optimal interferometric pair for DEMs should achieve both a sufficiently long perpendicular baseline for height accuracy and a short temporal baseline for coherence (Yu et al., 2019a). The Sentinel-1 constellation's 6-day repeat cycle (now restored with Sentinel-1C) is beneficial in this regard, as it keeps temporal baselines short; however, the relatively small orbital baselines of Sentinel-1 limit the vertical precision attainable from a single interferogram (Prats-Iraola et al., 2015). While the primary aim of Sentinel-1 was differential interferometry in the first place, the presented results show that any future SAR mission aimed at topographic mapping must carefully coordinate baseline geometry and revisit time to maximize DEM quality. One way to improve DEM accuracy even for Sentinel-1 data is by integrating multiple interferograms instead of relying on a single image pair. Recent research has shown that simple stacking of many InSAR DEMs can substantially reduce random errors (Ibarra et al., 2024).

It should be noted that only a small number of Sentinel-1A/C pairs from the experimental calibration phase met the geometric and quality criteria required for DEM generation in this study (22 frames within relative orbit #58 over Africa; see Figure 1), and after enforcing comparability of perpendicular baselines and environmental conditions, only a single frame remained eligible for the full 1/6/12-day comparison. Consequently, the presented accuracy estimates should be interpreted as site-specific and indicative, not as global performance metrics. While the methodology itself is established, broader generalization would require multiple frames per temporal baseline across diverse regions. In that sense, additional examples under similar geometrical conditions would be necessary to evaluate the sensitivity in the magnitude and pattern of error differences, but in this study, comparable perpendicular baselines across pairs were given priority to isolate the temporal-decorrelation effect on phase quality and elevation accuracy. The data scarcity is therefore a design consequence and reflects the reality of the brief calibration phase.

In summary, the presented findings have several implications for future SAR mission design and DEM generation strategies. The quality drop between 6 and 12 days aligns with the general known principles of radar interferometry that dense temporal

sampling is strongly beneficial for accurate DEM production, especially in environments prone to decorrelation (e.g. vegetated and urban areas) (Zebker and Villasenor, 1992). A return to or improvement upon the ~6-day repeat cycle (or even shorter) is worth pursuing to consistently achieve high coherence. At the same time, requirements for perpendicular baseline control come into play: mission designers should ensure a strategy that provides an optimal baseline distribution (neither too small to lose vertical precision nor too large to forfeit coherence). Upcoming SAR missions and enhancements (such as the combined use of C-band and L-band systems) can take these trade-offs into account. Ultimately, maintaining high coherence while maximizing elevation sensitivity will be key to improving DEM quality. By systematically isolating the temporal baseline effect, the presented results provide quantitative evidence to inform these future developments and the expected performance envelope of C-band InSAR for topographic mapping.

## 5. Conclusions

This study evaluated the quality of digital elevation models derived from Sentinel-1 interferometric pairs with temporal baselines of 1, 6 and 12 days, exploiting a short experimental acquisition phase of Sentinel-1C in early 2025. While the underlying methodology for DEM generation from Sentinel-1 data is well established, the uniqueness of this work lies in the availability of a true one-day repeat configuration under otherwise comparable acquisition geometry.

The results indicate that, for the investigated scene and acquisition period, DEMs derived from 1-day and 6-day interferometric pairs have comparable accuracy, whereas the DEM generated from a 12-day temporal baseline shows a substantially degraded performance. This degradation is reflected in increased noise levels, a pronounced vertical bias and a broader error distribution. The unexpectedly small difference between the 1-day and 6-day results, and the pronounced deterioration at 12 days, cannot be conclusively attributed to a single physical cause based on the available data. While factors such as temporal decorrelation, unwrapping errors, atmospheric phase contributions and surface moisture variability are likely contributors, the present dataset does not allow these effects to be disentangled quantitatively. In this case, the errors (RMSE and NMAD) of the 12-day DEM were three times larger than those of the 1-day and 6-day results, and a large systematic upward bias in elevations was observed. By separating systematic bias and large-scale trends from random elevation error, the bias analysis provides the key methodological insight of this study, enabling a robust interpretation of the experimental one-day repeat configuration. In terms of coherence, the 1-day and 6-day pairs benefited from the newly re-established 6-day orbit cycle of Sentinel-1A and 1C and retained higher coherence values, supporting the derivation of more reliable elevations, aligning well with the Copernicus reference DEM and ICESat-2 validation points.

The strict control of perpendicular baselines and environmental conditions to isolate temporal effects is a key strength of the study but also its main limitation. Due to the very restricted number of Sentinel-1C acquisitions during the short calibration phase, no systematic evaluation on error behaviour and impacts was possible. As a consequence, the presented results are scene-specific and season-specific, reflecting a particular combination of topography, land cover and environmental

conditions during the winter-early spring period. Different surfaces (e.g. dense forest, bare terrain in arid regions) or different atmospheric and phenological states may yield markedly different coherence behaviour and DEM accuracy.

In this context, the findings demonstrated of what can be achieved under exceptionally short temporal and large perpendicular baselines but cannot deliver causal interpretations or benchmarking for Sentinel-1 performance in general. Future work based on larger datasets, multiple regions and longer time series will be required to robustly quantify how DEM accuracy evolves with temporal baseline under varying environmental and geometric conditions.

**Code availability**

This article is not accompanied by code.

**Data availability**

This study contains modified Copernicus Sentinel data retrieved from the Copernicus Data Space Ecosystem in 2025 under the following product IDs:

- S1A_IW_SLC__1SDV_20250309T172310_20250309T172337_058230_07322D_22B9
- S1C_IW_SLC__1SDV_20250310T172151_20250310T172218_001381_0027A5_0B89
- S1A_IW_SLC__1SDV_20250414T172307_20250414T172334_058755_074751_83B1
- S1C_IW_SLC__1SDV_20250420T172155_20250420T172222_001979_003F75_9063
- S1A_IW_SLC__1SDV_20250426T172308_20250426T172335_058930_074E73_794B
- S1A_IW_SLC__1SDV_20250402T172307_20250402T172334_058580_074020_DB59

**Interactive computing environment**

No interactive computing environment was used in this article. *Please delete if appropriate.*

**Sample availability**

No registered samples were used in this article. *Please delete if appropriate.*

**Video supplement**

No video supplement is available. *Please delete if appropriate.*

## Supplements

No additional supplements are provided. *Please delete if appropriate.*

## Author contribution

AB developed the idea, processed and analyzed the data, prepared and revised the manuscript.

## Competing interests

The author declares that he has no conflict of interest.

## Acknowledgements

The author thanks the Open Access Publishing Fund of the University of Tübingen.

## Financial support

This study received no external funding.

## Review statement

*To be included by Copernicus.*

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

**Figures**

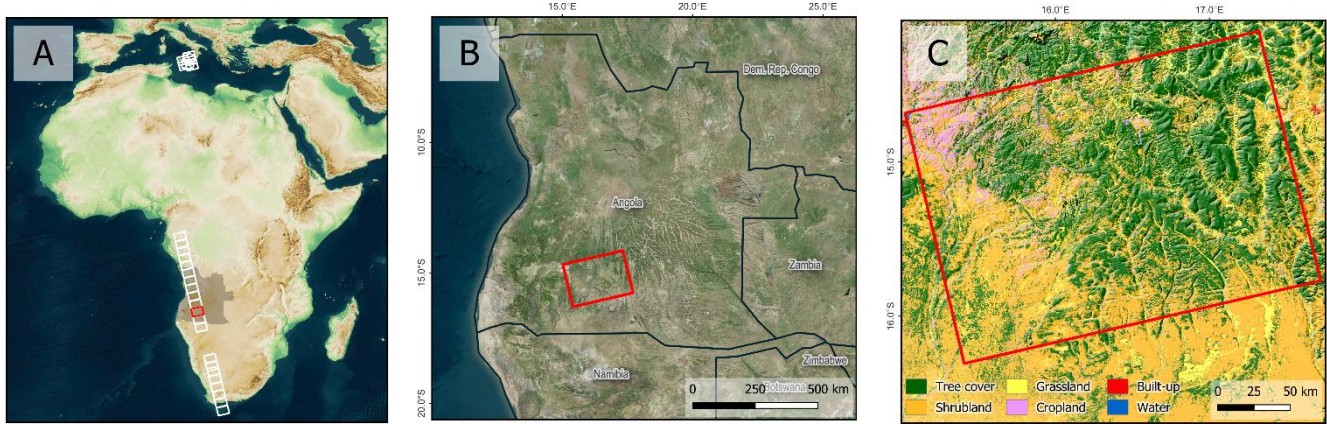

**Figure 1: Location of the selected frame within Africa (A) and within Angola (B), and land cover of the study area [ESA WorldCover] overlaid by DEM hillshade (C)**

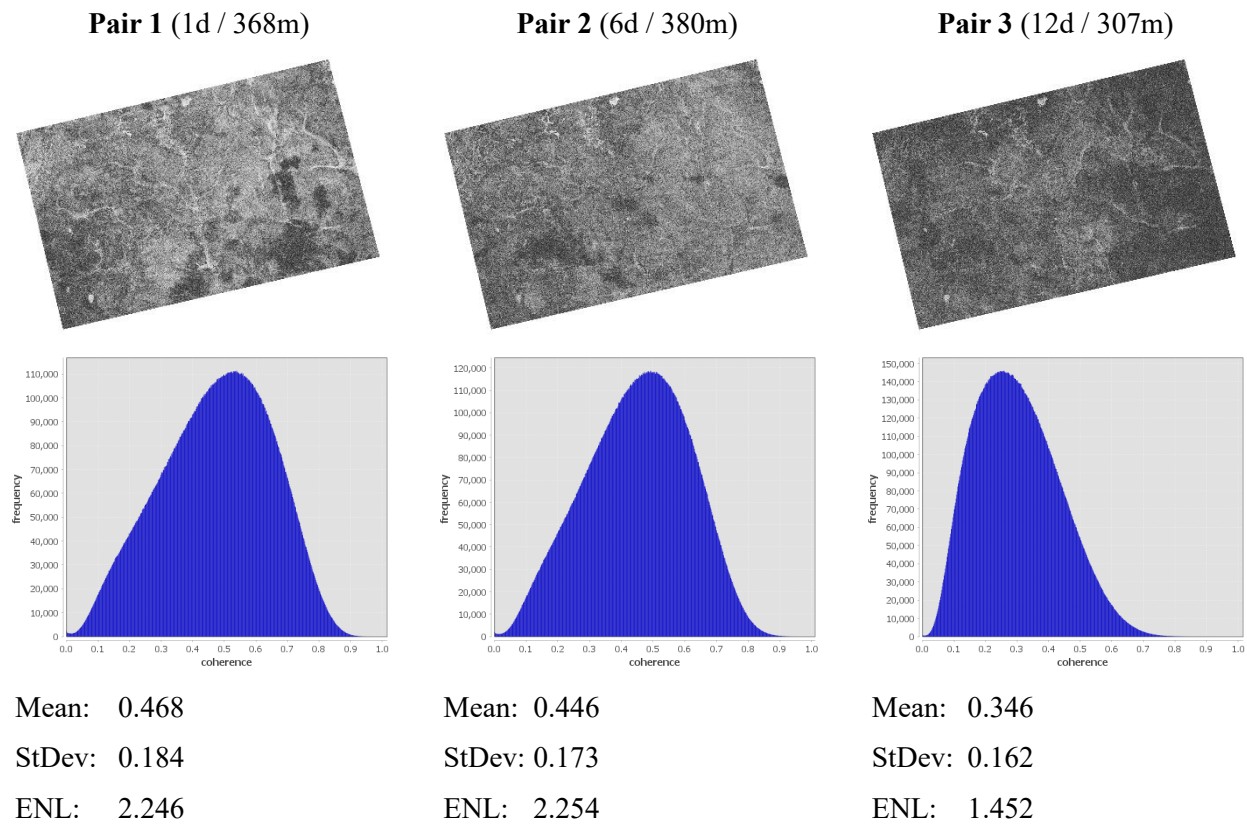

| **Pair 1** (1d / 368m) | **Pair 2** (6d / 380m) | **Pair 3** (12d / 307m) |
|---|---|---|
| Mean:  0.468 | Mean:  0.446 | Mean:  0.346 |
| StDev:  0.184 | StDev: 0.173 | StDev: 0.162 |
| ENL:  2.246 | ENL:  2.254 | ENL:  1.452 |

**Figure 2: Comparison of coherence for the three image pairs. Top: Map scaled between 0 (black) and 1 (white); Middle: Raster histogram; Bottom: Raster statistics with Mean=arithmetic mean, StDev=standard deviation, and ENL=equivalent number of looks.**

| Pair | Interferogram | Color-coded elevations |
|------|---------------|------------------------|

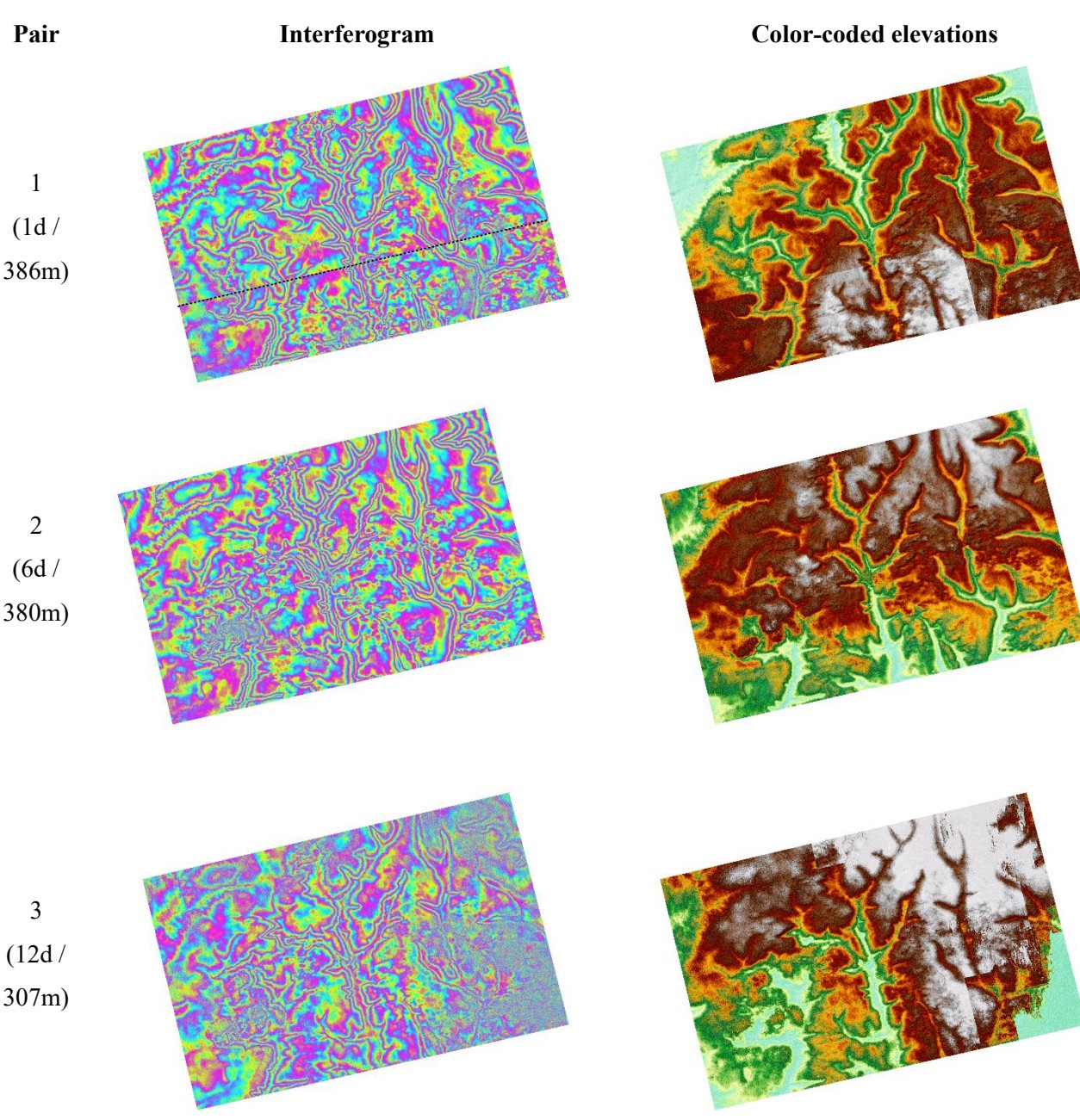

(1d /
386m)

(6d /
380m)

(12d /
307m)

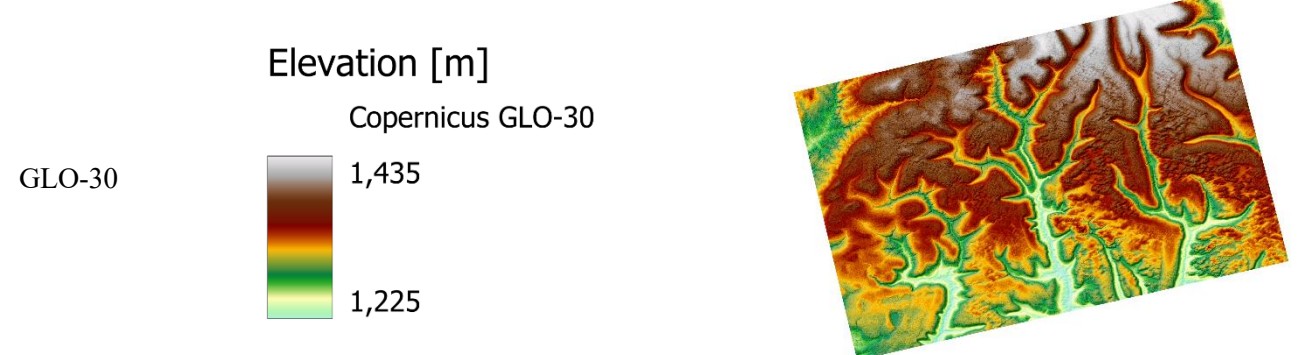

Elevation [m]
Copernicus GLO-30

GLO-30

1,435

1,225

**Figure 3: Interferograms (left) and resulting digital elevation model (right) for the three image pairs. Copernicus 30m Elevation Model (GLO-30) for visual reference.**

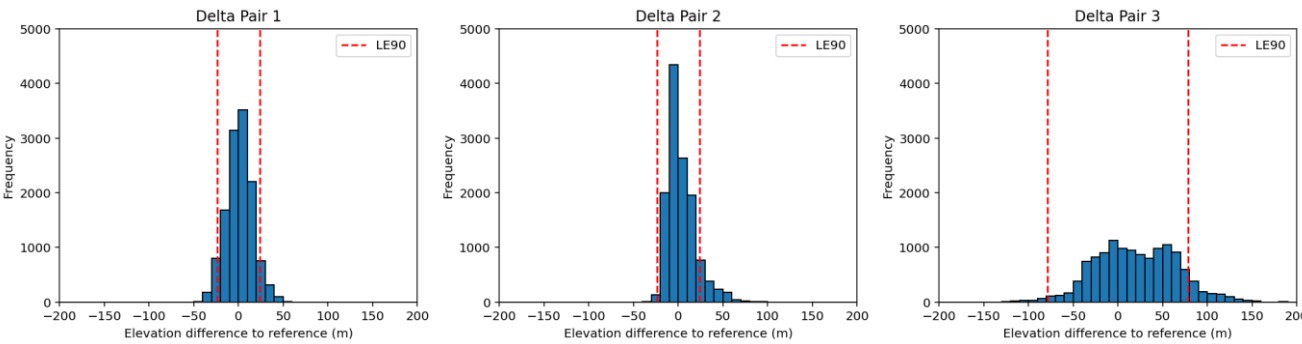

**Figure 4: Error histograms and LE90 range (dashed red line) of the three image pairs**

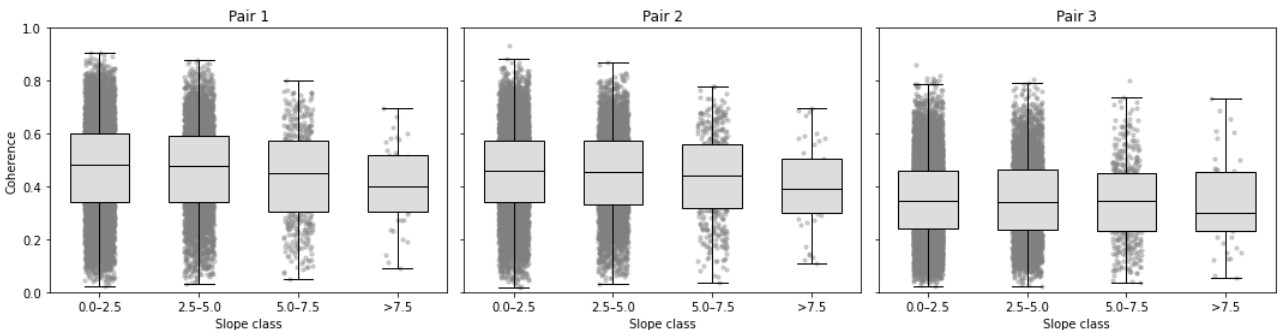

**Figure 5: Coherence of the analyzed image pairs disaggregated by terrain slope classes**

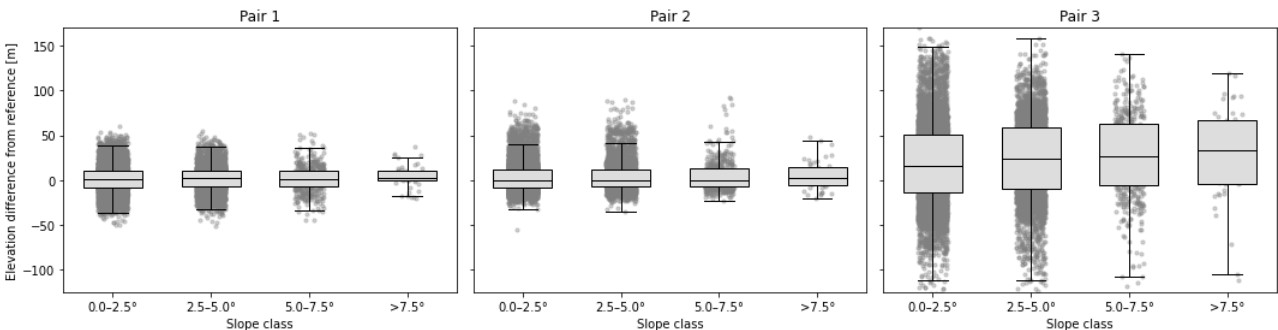

**Figure 6: Elevation differences of the analyzed image pairs disaggregated by terrain slope classes**

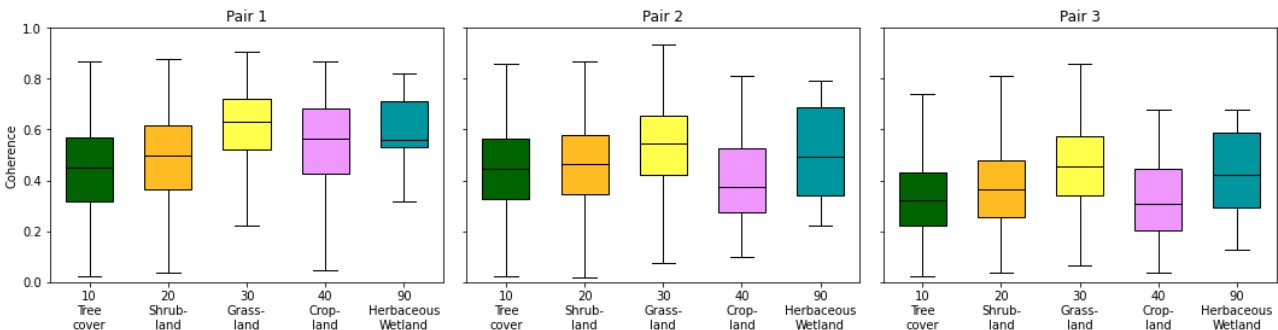

**Figure 7: Coherence of the analyzed image pairs disaggregated by land cover classes**

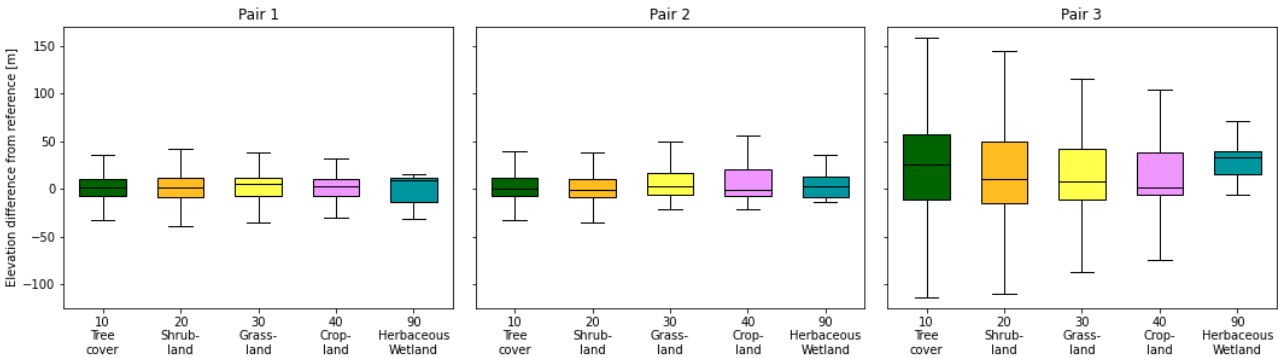

**Figure 8: Elevation differences of the analyzed image pairs disaggregated by land cover classes**

**Tables**

**Table 1: Interferometric pairs used in this study (satellites are indicated by superimposed letters A and C). Please note that only pairs 1 to 3 are intensively evaluated in this study and pair 4 was only computed to provide qualitative context for the 12-day result of pair 3.**

| Pair # | Reference | Support | $B_{temp}$ [d] | $B_{perp}$ [m] | HoA [m] |
|---|---|---|---|---|---|
| 1 | 09.03.2025[A] | 10.03.2025[C] | 1 | 386.3 | 39,53 |
| 2 | 14.04.2025[A] | 20.04.2025[C] | 6 | 380,5 | 40,18 |
| 3 | 14.04.2025[A] | 26.04.2025[A] | 12 | 307,1 | 49,81 |
| [4] | 14.04.2025[A] | 02.04.2025[A] | 12 | 147.9 | 103.34 |

**Table 2: Interferometric processing of radar image pairs**

| Name of the Process (SNAP) | Purpose | Parameters / comments |
| --- | --- | --- |
| TOPS Split | Selection of desired area and data configuration | VV polarization<br>Sub-swath 2, Bursts 2-4 |
| Apply Orbit File | Retrieval of precise orbit state vectors for enhanced positional accuracy (Fernández et al., 2024) | No orbit information was available for Sentinel-1C products |
| Back Geocoding | Coregistration of the reference and support product | Bilinear resampling<br>Supporting DEM: GLO-30 (ESA, 2022) |
| Enhanced Spectral Diversity | Estimation of azimuth and range offsets to increase coregistration quality within a network-based optimization process (Fattahi et al., 2017) | Registration window: 512x512<br>Search window: 16x16<br>Cross-correlation threshold: 0.1<br>ESD estimator: Periodogram |
| Interferogram Formation | Retrieval of interferometric phase and coherence of the image pair as raster images in slant range geometry | Subtraction of Flat-Earth Phase based on 501 points and a polynomial of degree 5<br>Coherence window size: 10x10 |
| Goldstein Phase Filtering | Improvement of interferogram quality by Fourier-based filtering (Goldstein and Werner, 1998) | FFT size: 64x64<br>Filter window size: 3x3<br>Coherence masking disabled |
| TOPS Deburst | Merging of bursts (2-4) in range direction based on time tags to remove seamlines | Seamline between burst 2 and 3 in pair 1 due to degraded calibration quality (Hajduch, 2025) |
| Phase Unwrapping | Translation of cyclic phase pattern into continuous measure along closed paths (Zebker and Lu, 1998) | Performed using the snaphu library (Zebker, 2020) outside SNAP |
| Phase to Elevation | Conversion of unwrapped phase into elevations of metric unit | Supporting DEM: GLO-30 (ESA, 2022) |
| Range Doppler Terrain Correction | Translation of the data from range geometry into a coordinate reference system (Curlander and MacDonough, 1991) | Supporting DEM: GLO-30 (ESA, 2022)<br>Bilinear resampling of DEM and radar image<br>Map projection: WGS84 (DD) |

**Table 3: Error metrics for the digital elevation models of the three image pairs analyzed in this study**

| Pair | RMSE [m] | NMAD [m] | LE90 [m] | Mean Bias [m] | MAPE [%] |
|------|----------|----------|----------|---------------|----------|
| 1 | 14.678 | 13.540 | 24.145 | 1.484 | 0.866 |
| 2 | 16.362 | 13.247 | 26.914 | 3.608 | 0.891 |
| 3 | 49.419 | 49.381 | 81.290 | 20.395 | 2.975 |

**Table 4: Error metrics for the digital elevation models of all three pairs before and after bias correction and ramp removal**

| Pair # | State | RMSE [m] | NMAD [m] | LE90 [m] |
|---|---|---|---|---|
| 1 | Raw (as in Table 3) | 14.97 | 14.97 | 14.97 |
|  | Bias-corrected | 13.92 | 13.92 | 13.92 |
|  | Bias and ramp removed | 14.79 | 14.79 | 14.79 |
| 2 | Raw (as in Table 3) | 15.95 | 15.95 | 15.95 |
|  | Bias-corrected | 15.94 | 15.94 | 15.94 |
|  | Bias and ramp removed | 15.86 | 15.86 | 15.86 |
| 3 | Raw (as in Table 3) | 49.3 | 49.4 | 81.3 |
|  | Bias-corrected | 57.9 | 50.1 | 71.1 |
|  | Bias and ramp removed | 48.2 | 22.4 | 51.2 |