# Peer review of "One-day repeat pass interferometry highlights the role of temporal baseline on digital elevation models retrieved from Sentinel-1"

_EGUsphere, 2025_

## Author Response (AR1)

**254266928 (Geo-spatial Information Science): "Urban structure types as reflections of socio-economic patterns. Revealing statistical links at the urban quarter scale.**

Dear anonymous reviewers,

I sincerely thank you for your careful reading of the manuscript and for the constructive and detailed comments provided. I highly appreciate the time and effort invested in evaluating my work and in offering thoughtful suggestions to improve its methodological rigor and clarity.

I have carefully considered all comments and have revised the manuscript accordingly. Wherever possible, I implemented your recommendations directly; where methodological or data-related constraints limited a full implementation, I aimed to address the underlying concerns through clearer justification, additional analyses, or explicit discussion. I believe that this process has substantially strengthened the manuscript in terms of transparency, methodological robustness, and interpretability.

Overall, I am confident that the revised version reflects a marked improvement over the original submission and more clearly communicates the scope, contributions, and limitations of the proposed approach. I hope that you reviewers will acknowledge these efforts and find that the revised manuscript satisfactorily addresses their concerns.

Please find below a detailed, point-by-point response to all reviewer comments, outlining the changes made and indicating where they can be found in the revised manuscript. I would like to thank the reviewers once again for their valuable feedback and the opportunity to revise and improve my work, and I appreciate the possibility to submit the manuscript for publication in this journal.

**Reviewer 1**

The paper describes the evaluation of a digital elevation model generated from three different pairs of Sentinel-1 acquisitions with a temporal lag of 1, 6 and 12 days. The results show, that the DEMs with 1 and 6 days temporal distance are comparable, while the DEM with 12 days is much worse. The paper is written very clearly and easy to read.

**General comments**

I understand, that there is only one feasible data set available to perform this analysis. However, a similar performance for 1 and 6 days temporal lag, and a such worse performance for 12 days is very unexpected in my opinion. The author does not provide a stringent explanation for this, but rather guessing, that it might be due to atmospheric effect (why should there be a dependency of seven days?) or whatever.

A more thorough error analysis analyzing the differences between the pairs is required to make this paper complete. In addition, an analysis comparing the performance for 6 days and 12 days repeat with similar measures for Sentinel-1A versus Sentinel-1B (best over the same area) should be easily possible and would confirm these massive performance deterioration after 12 days.

In my opinion, the conclusion (12 days very bad, 1 and 6 days good -> I really would also expect some differences between 1 and 6 days) is very miss-leading for designer of future interferometric missions.

→ *Thank you for raising these aspects. The similarity between the 1-day and 6-day results, combined with the strong performance degradation at 12 days, was unexpected to me as well. I therefore carefully reassessed both the interpretation and the robustness of this finding, as described later.*
*As outlined in the manuscript, the analysis is constrained by the very specific acquisition conditions of the Sentinel-1C calibration phase, during which only a small number of frames were acquired with unusually large perpendicular baselines (>300 m). Within this short time window, only one interferometric frame worldwide fulfilled all criteria required for a controlled comparison of 1-, 6- and 12-day temporal baselines while keeping perpendicular baseline, topography and land cover comparable. Consequently, no additional pairs were available that would allow a systematic repetition of the experiment under equivalent geometric conditions.*

→ *I nevertheless fully agree that a deeper error analysis is required, and I therefore extended the study as follows. Section 3.4 (Bias analysis) was added and contains a detailed error decomposition. After separating systematic bias and analysing the spatial structure of the residuals, a pronounced planar ramp in the 12-day pair was identified, with gradient magnitudes an order of magnitude larger than in the 1- and 6-day pairs. Removing this ramp reduces the NMAD by more than 50 % and substantially lowers LE90, indicating that the observed degradation is dominated by pair-specific long-wavelength errors rather than by the temporal baseline itself. The manuscript has been revised accordingly to reflect this finding and to avoid any generalised interpretation of the 12-day result.*

→ *Second, following your suggestion, I analyzed an additional interferometric pair with a 12-day temporal baseline from this period data. Although this supplementary pair exhibits a substantially smaller perpendicular baseline (~150 m) and is therefore not directly comparable in terms of height sensitivity, it provides qualitative context for long temporal baselines. To avoid diluting the main analysis, this comparison is presented as a supplementary evaluation attached to this review and explicitly separated from the core results. The comparison shows clearly different error characteristics, particularly with respect to bias magnitude and dispersion. This lack*

*of consistency indicates that the strong degradation observed in the initial 12-day pair (pair 3 in the manuscript) cannot be attributed to the temporal baseline alone. Instead, it points to pair-specific error propagation related to acquisition geometry and phase referencing, rather than to a systematic threshold effect between 6- and 12-day repeat intervals.*

→ *Based on these additional analyses, I revised the manuscript to remove speculative causal statements and to clarify that the reported behavior should be interpreted as descriptive and scene-specific. The conclusions were entirely rephrased and extended accordingly to emphasize that the results do not support a generalized statement about 12-day interferometric performance, but rather document an empirical outcome enabled by a unique calibration-phase configuration.*

→ *If you think, based on these changes, the title should be changed as well, please communicate.*

I would recommend a major revision for this paper.

**Specific comments**

Line 101: Please specify the 14 m wide ground track in more detail. Data points with a diameter of 14 m? Distance in along track?

→ *The description was specified and a reference was added to make it clearer.*

Figure 3: The elevation value in the legend should be 1,225 m and 1,435 m correct? It seems, that there is a large ramp in flight direction on pair 1. Could this be calibrated to improve the results?

→ *Yes, the thousands separator was now changed to comma. The ramp in flight direction is probably caused by unwrapping errors resulting from the seamline artifact between the two bottom bursts (see Table 2). This is caused by the less rigorous calibration of the S1C data during that time and I was not successful correcting it during the Debursting step.*

Line 190: The analysis on these effects should be intensified as they drive the conclusion.

→ *Please see the general response above and the provided error analysis. The conclusions were adjusted accordingly to avoid claiming a general explanation on the impact of temporal baseline.*

Line 212ff: What about shadow/layover regions? How are these considered?

→ *I have not specifically addressed shadow/layover regions in the error analysis because slopes > 39° (incidence angle) (indicating layover regions) and of slopes > 51° (90-incidence angle) are rarely available in the study area (90% of all pixels are below 5°) in the moderate terrain of the study area. This was added in the description of the study area. These effects may still occur locally on very steep slopes and could contribute to the observed outliers. I added this in the text.*

Line 276/277: "This is consistent with previous studies…": I could not find a satisfactory consistent analysis considering the importance of the temporal baseline of < one week in (Wu and Madson, 2024) except the reference to (Braun, 2021). In my opinion it would be really important to cite independent studies analyzing the difference between 6 and 12 days Sentinel-1 interferograms/DEMs.

→ *Wu and Madson (2024) were added as a overall reference to demonstrate general factors influencing DEM quality. After further research, I was able to add at least*

*three studies that compare six-day pairs with higher baselines in order to provide figures here as well.*

Line 290: "correcting the 20 m bias… would bring its accuracy considerably closer to the shorter-baseline results.": Please perform this correction and re-evaluate the results.

→ *This has been done as outlined above and added as a new section (3.4 Bias analysis)*

Line 323: "additional examples under similar geometrical conditions would have likely refined the magnitude of error differences but not the central pattern observed here": I would definitely disagree with this statement. The author has only assessed one single data set. This can be deteriorated by whatever. To conclude, that the "central pattern" would be valid for most other measurements is really mis-leading in my opinion.

→ *Fully agree here. The wording was too strong and implied generality that is not supported in the data. In the light of the new findings, I have removed this claim entirely and replaced it with a more cautious formulation. The revised text now explicitly states that additional examples would primarily serve to assess the variability and robustness of the observed behaviour and that no inference about general or typical error patterns is possible based on the present dataset.*

**Minor comments / editorials**

Line 31: Sentinel-1 instead of "Sentinel 1"

→ *corrected*

Line 57: Abbreviation IW not introduced

→ *added the full term, thank you for pointing it out*

Line 110: "Figure 2 <space> shows"

→ *corrected*

Line 125: "Comparison of coherence"

→ *corrected*

Line 133: "at the bottom. Pair 1"

→ *corrected*

Line 143: German data format

→ *all dates in the text are now consistently formatted as required in the instructions, e.g. 25 July 2007 (dd month yyyy)*

Line 144: "surface. Pair 3"

→ *corrected*

Figure 4: All histograms from "Delta Pair 3" as the title claims?

→ *Thank you for noticing, the histogram titles have been updated (Delta Pair 1, Delta Pair 2, Delta Pair 3)*

All over: Percentages should be uniquely written (50 % vs. 50%)

→ *I now consistently removed all spaces between numbers and percentages*

Line 239: ". ThesE were retrieved…"

→ *corrected*

Line 244: "grassland" twice

→ *removed*

Line 365: Capital letters for "Forests, Shrublands, Wetlands"

→ *All class names now consistently start with capital letters*

Line 306: This instead of "Tis"

→ *corrected*

**Reviewer 2**

The study computes and analyses three radar interferograms (1, 6, 12 day temp. baseline) based on Sentinel-1C calibration phase data for their suitability for generation of digital elevation models. The study is timely, mainly due to the new data type acquired during the Sentinel-1C calibration phase. The conclusions of the study confirm in general known principles and experiences from radar interferometry. I suggest the author considers the below comments, mostly minor, except the last one that I consider more major:

There are a number of typos that use of a spell-checker should have indicated. This and some other small mistakes suggest the paper was submitted in a rush. I feel referees deserve a better checked version of a paper submission.

→ *You are right, the paper contained several typos and slips. Highlighting these should not be the duty of the reviewers. I carefully checked and corrected the entire manuscript during the revision.*

Line 37: I'd say that not only temporal baselines limit topographic InSAR, but equally also the design of spatial (i.e. perp.) baselines.

→ *I added this aspect and a reference to the small orbital tube of the S1 mission.*

41: "various studies". References?

→ *The references are included in the review paper at the end of the sentence.*

47: ... short temporal baselines ...

→ *"temporal" was added in the sentence.*

74: Some specifications of the study site topography would be very useful: min, max, avg., elevations, slopes, etc.

→ *Descriptions of the topography were added in chapter 2.*

Fig. 3: lower left legend "1.435", "1.225": unit?

→ *The thousands separator was now changed to comma, the unit [m] is present in the heading.*

Fig. 3: I don't think the right panels are hillshades? Rather colour-coded elevations?

→ *You are fully right, the heading was changed.*

Fig. 3: the difference between the colour-coded elevations of pair 1 and reference look the largest among all three, yet the error specs are smallest. I don't understand. Was there an additional adjustment step applied?

→ *This is explained in line 147f (now additionally added to Table 2). Due to the degraded metadata quality, there was a seamline between bursts 2 and 3 which*

*could not be removed during the debursting step, introducing further uncertainty. It is indicated by a dashed line in the corresponding interferogram. I have additionally addressed this as a factor in the discussion.*

Fig. 4: the panel titles refer all to pair 3. What about pairs 1 and 2?

→ *Thank you for noticing, the histogram titles have been updated (Delta Pair 1, Delta Pair 2, Delta Pair 3)*

around 287: how was the vertical reference during the unwrapping fixed? Average to the reference DEM? Or a reference point? Or else? In any case, where does then a vertical bias stem from?

→ *As indicated in Table 2, vertical reference is provided by the Copernicus DEM (GLO-30) during the conversion of the unwrapped signal into heights by the SNAP operator "Phase to Elevation". The vertical bias is now additionally analyzed in a new section "3.4. Bias analysis" (as recommended by Reviewer 1), indicating that there is a strong ramp with gradient magnitudes of approximately 0.90 m/km in East–West direction and 0.98 m/km in North–South direction. The article now clearer separates the indications for a superimposed bias (e.g. by residual orbital errors, large-scale atmospheric phase contributions, or said unwrapping errors) and those who can be attributed to longer temporal baselines.*

338, Conclusions: this section should clearly state (again) the limitation of the study that it refers to one site and time, with specific topography and land cover, for a specific season and vegetation state. Results could look quite different for other types of surfaces, e.g. arid land, or dense forest. It is also not clear to what extent the large coherence loss between 6 and 12 days is typical for the site, season and vegetation, or just a random event (weather, atmosphere, vegetation, acquisition, ...). The conclusions sound like a systematic study, but actually (by necessity of the available data) it is not. The conclusions should be less general and wide as this is not supported by the specific study. At least investigation of the variation of coherence over time doesn't require Sentinel-1A and -1C pairs and could be used to better understand the found coherence deterioration.

→ *Thank you for your explanations, I fully agree. This was also criticized by the other reviewer, so another chapter ""3.4. Bias analysis" was added and conclusions were entirely revised and extended to make them less general and more transparent about the data- and site-dependency of the results. The study is again characterized as an empirical case study, less a systematic evaluation.*

**Error analysis of digital elevation model quality of two Sentinel-1 images of 12 day temporal baseline**

Complementing the PrePrint "One-day repeat pass interferometry highlights the role of temporal baseline on digital elevation models retrieved from Sentinel-1"

https://doi.org/10.5194/egusphere-2025-3998

Andreas Braun

09.01.2026

**Image IDs**

S1A_IW_SLC__1SDV_20250414T172307_20250414T172334_058755_074751_83B1

S1C_IW_SLC__1SDV_20250420T172155_20250420T172222_001979_003F75_9063

S1A_IW_SLC__1SDV_20250426T172308_20250426T172335_058930_074E73_794B

S1A_IW_SLC__1SDV_20250402T172307_20250402T172334_058580_074020_DB59

**Image Pairs**

| Pair # | Reference | Support | $B_{temp}$ [d] | $B_{perp}$ [m] | HoA [m] |
|---|---|---|---|---|---|
| 3 | 14.04.2025 | 26.04.2025 | 12 | 307.1 | 49.81 |
| 4 | 14.04.2025 | 02.04.2025 | 12 | 147.9 | 103.34 |

$B_{temp}$   = temporal baseline

$B_{perp}$   = perpendicular baseline

 HoA   = Height of ambiguity

For processing parameters, please refer to the manuscript

Elevation Model (hillshade) of Copernicus-DEM (GLO-30) for visual reference

[Figure]

**Visual Comparison**

| Pair 3 | Pair 4 |
|:---:|:---:|

**Coherence**

[Figure]

**Interferogram**

[Figure]

**Elevation Model (hillshade)**

[Figure]

**Coherence**

[Figure]

**Interferogram**

[Figure]

**Elevation Model (hillshade)**

[Figure]

**Error Measures**

| Pair 3 | Pair 4 |
|:---:|:---:|
| Scatter Plot | Scatter Plot |

[Figure]

[Figure]

| | |
|:---:|:---:|
| RMSE: 49.419 m | RMSE: 37.414 m |
| NMAD: 49.381 m | NMAD: 57.674 m |
| Mean Bias: 20.395 m | Mean Bias: 15.683 m |